# Partitioning the Effects of Soil Legacy and Pathogen Exposure Determining Soil Suppressiveness via Induced Systemic Resistance

**DOI:** 10.3390/plants11212816

**Published:** 2022-10-23

**Authors:** Na Zhang, Chengzhi Zhu, Zongzhuan Shen, Chengyuan Tao, Yannan Ou, Rong Li, Xuhui Deng, Qirong Shen, Francisco Dini-Andreote

**Affiliations:** 1Jiangsu Provincial Key Lab of Solid Organic Waste Utilization, Jiangsu Collaborative Innovation Center of Solid Organic Wastes, Educational Ministry Engineering Center of Resource-Saving Fertilizers, Nanjing Agricultural University, Nanjing 210095, China; 2The Key Laboratory of Plant Immunity, Nanjing Agricultural University, Nanjing 210095, China; 3Department of Plant Science, The Pennsylvania State University, University Park, PA 16802, USA; 4Huck Institutes of the Life Sciences, The Pennsylvania State University, University Park, PA 16802, USA

**Keywords:** bacterial wilt, suppressive soil, split-root system, soil legacy, transcriptome, plant systemic resistance

## Abstract

Beneficial host-associated bacteria can assist plant protection against pathogens. In particular, specific microbes are able to induce plant systemic resistance. However, it remains largely elusive which specific microbial taxa and functions trigger plant immune responses associated with disease suppression. Here, we experimentally studied this by setting up two independent microcosm experiments that differed in the time at which plants were exposed to the pathogen and the soil legacy (i.e., soils with historically suppressive or conducive). Overall, we found soil legacy effects to have a major influence on disease suppression irrespective of the time prior to pathogen exposure. Rhizosphere bacterial communities of tomato plants were significantly different between the two soils, with potential beneficial strains occurring at higher relative abundances in the suppressive soil. Root transcriptome analysis revealed the soil legacy to induce differences in gene expression, most importantly, genes involved in the pathway of phenylpropanoid biosynthesis. Last, we found genes in the phenylpropanoid biosynthesis pathway to correlate with specific microbial taxa, including *Gp6*, *Actinomarinicola*, *Niastella*, *Phaeodactylibacter*, *Longimicrobium*, *Bythopirellula*, *Brevundimonas*, *Ferruginivarius*, *Kushneria*, *Methylomarinovum*, *Pseudolabrys*, *Sphingobium*, *Sphingomonas*, and *Alterococcus*. Taken together, our study points to the potential regulation of plant systemic resistance by specific microbial taxa, and the importance of soil legacy on disease incidence and eliciting plant-defense mechanisms.

## 1. Introduction

The structure and composition of plant-associated microbiomes are tightly connected with plant growth, health, and performance [1,2,3]. In this sense, the support of plant health and productivity is a global concern for food safety and security [4,5,6]. It is possible that this can be achieved by better understanding how plant roots dynamically interact with soil microorganisms [7]. For instance, the phenomenon of disease suppression mediated by soil microbes has long been described for a variety of soilborne pathogens in different areas across the globe [7,8,9,10]. Thus, it is plausible to assume that properly understanding the biological mechanisms underpinning the status of suppressive soils offers an opportunity to advance the targeted and effective manipulation of beneficial microbiomes.

The occurrence and impact of soilborne diseases are related to host susceptibility, pathogen presence and infection, and the biotic and abiotic environment, including the soil and plant-associated microbiomes [11]. The plant rhizosphere can be seen as a soil hotspot for microbial activity and biological interactions [7,12]. Such an interactive system is dynamically affected by plant root phenology, root exudates, and other carbon-derived compounds released by the plant roots (e.g., mucilage, cell lysates, etc.) [13,14] that collectively influence plant health and immunity [15]. In this context, it has been described the important role of specific microbial taxa and functions directly affecting plant growth and health. For instance, it was shown that upon pathogen exposure, plants can direct the recruitment of beneficial/protective microbial taxa from the soil to enhance or promote disease suppression [12,16]. In another example, it was found that specific *Pseudomonas* spp. can produce the antifungal metabolite 2,4-diacetylphloroglucinol that acts as chemical control of the pathogen resulting in disease suppression [17].

Most studies on disease-suppressive soils have been focused on reporting the direct antagonistic effect of protective bacteria in the rhizosphere by inhibiting the growth of specific pathogens [18,19]. Besides, some studies have also reported the status of soil suppressiveness to be associated with an indirect mechanism in the plant, that is, via induced systemic resistance, ISR [20,21]. For instance, the microbial activation of jasmonic acid (JA) and ethylene (ET)-dependent elicitation were recently reported in Arabidopsis and tomato plants [22,23]. In line with these studies, we here hypothesized that the soil legacy effects (i.e., history of disease suppression; from suppressive to conducive soils) can lead to different compositions in the tomato rhizosphere microbiome. This might have direct implications for the level of disease suppressiveness—in this case—as mediated by the ability of specific taxa to elicit ISR against the pathogen *Ralstonia solanacearum* (Smith) Yabuuchi et al. emend. Safni et al. To test this hypothesis, we collected soils in a long-term field experiment cultivated with tomato and displaying different levels of infection by the targeted pathogen. We performed two independent experiments based on a tomato split-root system to test (1) variation in disease incidence; and (2) the interactive effect of disease incidence based on soil legacy effects and time of exposure to the soilborne pathogen (0 and 7 days after planting). Changes in the tomato rhizosphere microbiome and disease incidence were analyzed in line with the ability of taxa to induce systemic resistance and based on the enrichment of potential functional pathways in plants cultivated in disease-suppressive soil systems.

## 2. Materials and Methods

### 2.1. Soil Sample Collection

Soils with different histories (i.e., suppressive and conducive soils, respectively) were collected from a long-term experimental field site located at the Nanjing Institute of Vegetable Science, Nanjing, China (32°02′ N, 118°50′ E). The disease suppressiveness of the suppressive soil was determined in our previous study [24]. In brief, conducive soils were mixed with 10% of suppressive soils heat-treated at different temperatures. Then, the disease suppressiveness against the pathogen *R. solanacearum* was evaluated via seedling disease incidence. At this site, tomato plants have been cultivated as a single crop for several seasons. The suppressive soil was obtained from a field treatment that has been treated with bio-organic fertilizer, defined as “BF”. This treatment resulted in an average bacterial wilt incidence around 30% in Spring seasons. The conducive soil was obtained from a field treatment that has been amended with chemical fertilizer, defined as “CF”. This treatment resulted in an average bacterial wilt incidence around 75% in Spring seasons. In both of these sites, soil samples were collected and transferred to the laboratory (<24 h). For soil sample collection, soil cores at depths of 0–15 cm were used. Soil processing consisted of air drying, homogenization, and sieving to remove plant debris. After that, half of the soils were stored in plastic bags, and the other half was subjected to sterilization by gamma radiation (60 KGy).

### 2.2. Culturing Aseptic Tomato Seedings

*Solanum lycopersicum* cv. ‘Hezuo 903’ tomato, *Solanum lycopersicum* L. (a commercial cultivar) was used as a susceptible plant. Seeds were surface-sterilized using 70% ethanol for 10 min, 100% ethanol for 1 min, and 3% NaClO for 1 min [25], transferred to sterilized Murashige and Skoog (MS, supplemented with 2% sucrose) semi-solid medium, and incubated at 25 °C for 3 days in the dark [25,26]. After germination and the emergence of primary roots, plates containing these seeds were placed in a greenhouse at a photoperiod of 16 h of light at 28 °C and 8 h of dark at 25 °C for 10 days. The obtained seedlings were transferred to Erlenmeyer flasks filled with sterilized water to properly establish the root system. Plants with fully established roots (~7 days) were transferred to split-root system pots for further experimentation.

### 2.3. Experimental Design

The pathogen *R. solanacearum* strain QL-Rs1115, isolated from the rhizosphere of a wilted plant in an infested field site in Qilin Town (118°57′ E, 32°03′ N, Nanjing city, China), was cultured at 30 °C with 170 rpm in liquid NB medium (0.5 g of yeast extract, 3.0 g of beef extract, 5.0 g of peptone, and 10.0 g of glucose per liter) for 36 h [27]. The root system of tomato seedlings was divided equally into two parts (boxes) in the split-root system. Each box consisted of a 250 mL glass box (as shown in Figure 1). In brief, the two boxes were set as follows: (i) the right side was filled with sterilized BF soil, and the left side was filled with natural BF soil, (ii) the right side was filled with sterilized CF soil, and the left side was filled with natural CF soil. Each box was filled with 200 g of the respective soil. This design allowed for two independent experiments to be performed, as follows: Experiment I. *R. solanacearum* QLRs-1115 was introduced into the soil (at a concentration of 10^6^ cells g^−1^ soil) in the right (sterilized) side of the box at the same time that plants were transferred to the split-root system. For this experiment, samples in the left (natural) side of the box were collected 7 days after the transplant. Samples were defined as “S”, which contained four treatments: SBF (rhizosphere derived from the BF soil without *R. solanacearum* inoculation), SCF (rhizosphere derived from the CF soil without *R. solanacearum* inoculation), SBF-Rs (rhizosphere derived from the BF soil and inoculated with *R. solanacearum* at the same time that plants were transferred to the split-root system), SCF-Rs (rhizosphere derived from the CF soil and inoculated with *R. solanacearum* at the same time that plants were transferred to the split-root system) (Figure 1a). Experiment II. Tomato plants were set to grow in the split-root system for 7 days prior to the introduction of *R. solanacearum* QL-Rs1115 (at a concentration of 10^6^ cells g^−1^ soil) into the soil in the right (sterilized) side of the box. For this experiment, samples on the left (natural) side of the box were collected 15 days after the transplant. Samples were defined as “E”, which contained four treatments: EBF (rhizosphere derived from the BF soil without *R. solanacearum* inoculation), ECF (rhizosphere derived from the CF soil without *R. solanacearum* inoculation), EBF-Rs (rhizosphere and root derived from the BF soil and inoculated with *R. solanacearum* 7 days after the plant establishment in the split-root system), ECF-Rs (rhizosphere and root derived from the CF soil and inoculated with *R. solanacearum* 7 days after the plant establishment in the split-root system) (Figure 1b). Each split-root system unit consisted of 3 plants, and each treatment contained 7 replicates (total of 21 plants per treatment). All replicates were randomly placed and cultured in a 28 ℃ incubator with constant moisture (25% of field capacity). The bacterial wilt disease incidence was recorded based on a bioassay including leaf wilting, leaf necrosis, and the whole plant wilted or dead, and calculated as the percentage of plants with bacterial wilt based on the total number of plants in each treatment.

### 2.4. Rhizosphere Sample Collection

Rhizosphere samples and roots of tomato seedlings growing on the left side (natural soil without *R. solanacearum* QL-Rs1115 inoculation) of the root box were collected in each treatment. Rhizosphere samples were collected as previously reported [28]. In brief, the root system was entirely removed, shaken vigorously to remove excess soil, and the soil tightly adhering to the roots was considered rhizosphere samples. Samples were transferred into a 50 mL centrifuge tube containing 20 mL of sterile phosphate saline buffer and shaken for 30 min at 170 rpm to separate the soil from the roots. After that, the roots were removed and the tubes were centrifuged to recover the rhizosphere sample. Rhizosphere samples were collected using six samples in each treatment, and the mass of each sample was greater than 0.5 g. All rhizosphere samples were stored at −80 °C for further DNA extraction. For root sample collection, root sections of each plant were harvested, immediately frozen in liquid nitrogen, and stored at −80 °C for further RNA extraction. Root samples were collected using three samples in each treatment, and the mass of each sample was greater than 2.0 g.

### 2.5. Soil DNA Extraction and Soil Microbial Abundance Determination

Soil DNA was extracted from 0.5 g of rhizosphere sample using the Power Soil DNA isolation kit (MOBIO Laboratories, Inc., Carlsbad, CA, USA), according to the manufacturer’s instructions. The quality and concentration of DNA in each sample was determined using the NanoDrop 2000 spectrophotometer (Thermo Scientific, Waltham, MA, USA), based on the 260/280 nm and 260/230 nm absorbance ratios. The abundance of *R. solanacearum* was determined by quantitative PCR (qPCR) on an Applied Biosystems StepOne Plus (Applied Biosystems, Foster City, CA, USA). The primer set used specifically target the pathogen *R. solanacearum* (*FlicF* and *FlicR*) [29]. Standard curves were generated using 10-fold serial dilutions of a plasmid containing the *fliC* gene of *R. solanacearum*. The qPCR assay was performed in a 20 μL reaction containing 10μL of SYBR Premix Ex Taq (2×, TaKaRa Bio Inc., Kusatsu-shi, Japan), 1 μL of template DNA (20 ng/μL), 1 μL of each forward and reverse primer (10 mM), and 7 μL of sterile water. For each sample three technical replicates were performed. Results are expressed as log10 values (copies/g soil), and used for further statistical analysis.

### 2.6. Illumina Miseq Sequencing

The total DNA obtained from the rhizosphere samples (Experiment I: after 7 days, Experiment II: after 15 days) in both experiments were subjected to amplification and sequencing of the bacterial 16S rRNA gene on an Illumina MiSeq PE250 platform at Personal Biotechnology Co., Ltd. (Shanghai, China). The partial bacterial 16S rRNA gene was amplified using primer set 520F (5′-AYT GGG YDT AAA GNG-3′) and 802R (5′-TAC NVG ATC TAA TCC-3′), containing specific Illumina adaptor for further library preparation and sequencing. All raw data were uploaded onto NCBI SRA under the sample accession number PRJNA824941.

### 2.7. Bioinformatics Analysis

Raw sequences were processed using the UPARSE pipeline [30]. Briefly, sequence quality was filtered with a maximum expected error of 0.5 and a length greater than 200 bp. After discarding sequence errors and singletons, the remaining sequences were assigned to OTUs with a threshold of 97% of nucleotide identity. Bacterial representative sequences were searched through the RDP classifier against the RDP Bacterial 16S rRNA gene database [31]. Mitochondrial and chloroplast sequences were removed using USEARCH. To obtain an equivalent sequencing depth, samples were rarified to an equal number of sequences of 29,795 (Experiment I), and 32,959 (Experiment II) in R using the “GUniFrac” package (function: rarefy). *Alpha*-diversity indices including Chao1 richness and Shannon diversity were determined using “vegan” (function: diversity). Non-metric multidimensional scaling (NMDS) based on a Bray–Curtis dissimilarity matrix was performed using “vegan” to explore differences between treatments. Permutational multivariate analysis of variance (PERMANOVA) was used to test the significance of community differences using “vegan” (function: adonis) with 9999 permutations. The relative abundance of each genus was calculated, then the Spearman’s rank correlation coefficients between *R. solanacearum* and other genera were conducted to count the groups significantly associated with *R. solanacearum* using “vegan”. Differential abundance analysis of OTUs across treatments was based on OTUs-relative abundances using the Linear discriminant analysis effect size (LEfSe). To do so, the Kruskal–Wallis (KW) test was used to detect OTUs that significantly differed between treatments based on 95% confidence intervals, and the linear discriminant analysis (LDA) was performed to estimate the effect size (LDA ≥ 2) [32]. The Mantel test was implemented in R “vegan” package to identify the correlation between bacterial communities with relative abundance values of *R. solanacearum* or the phenylpropanoid biosynthesis pathway. Fast tree and muscle software were used to construct the phylogenetic tree of OTUs (https://itol.embl.de/itol.cgi, accessed on 10 May 2020). To evaluate the interaction of different OTUs and DEGs (differentially expressed genes), co-occurrence networks were constructed based on Spearman’s correlation. This was calculated for all possible correlations between OTU × OTU, OTU × gene, and gene × gene. A robust co-occurrence was determined with a Spearman’s correlation coefficient (*r*) > 0.6 and statistically significant (*p*-value < 0.05) [33,34]. Network analyses were carried out using R “Hmisc” and “igraph” packages. Network visualization was conducted using Cytoscape version 3.7.0.

### 2.8. RNA-Sequencing (RNA-seq) and Transcriptome Analysis

Root samples from Experiment II collected after 15 days were subjected to total RNA extraction, cDNA preparation, and transcriptome sequencing at Personal Biotechnology Co., Ltd., Shanghai, China. In brief, total RNA was extracted using the RNAout kit (TIANDZ, CAT#: 71203) for RNA-Seq analysis, and using TRIZOL (Invitrogen) for qPCR analysis. RNA quality was determined using an Agilent 2100 Bioanalyzer (Agilent Technologies) and checked using RNase-free agarose gel electrophoresis. The cDNA libraries were constructed using an Agilent High Sensitivity DNA Kit, with a fragment length of 300–400 bp. The library was sequenced on an Illumina NextSeq 500 platform (150 bp paired-end reads). For each sample, raw reads were filtered using Cutadapt (Version 1.2.1) to obtain high-quality reads. The parameters/steps used were: (i) removal of adaptor reads; (ii) removal of low-quality reads with average quality score Q < 20; (iii) removal of reads with length < 50 bp. The data quality of clean reads was assessed using FastQC (http://www.bioinformatics.babraham.ac.uk/projects/fastqc, accessed on 2 March 2017). A total of 36 million raw reads were obtained for each sample, resulting in ~98% of quality-filtered reads (Appendix A). High-quality reads were mapped to the reference genome based on Ensembl (http://www.ensembl.org, accessed on 2 March 2017), and the reference genome used was Solanum_lycopersicum.SL2.50.dna.toplevel.fa. Read counts were obtained using HTSeq 0.6.1p2 (http://www-huber.embl.de/users/anders/HTSeq accessed on 2 March 2017), and subjected to read mapping counts using RPKM (Reads Per Kilobase per Million reads). The differential expression analysis was carried out using DESeq (version 1.18.0, http://www.bioconductor.org/packages/release/bioc/html/DESeq.html, accessed on 2 March 2017), with the condition of |log2fold-change| > 1.0 and *p*-value < 0.05, for significant changes in gene expression between the two treatments. The enrichment analysis was annotated based on GO (Gene Ontology, http://www.geneontology.org accessed on 6 March 2017) and KEGG (Kyoto Encyclopedia of Genes and Genomes, http://www.genome.jp/kegg, accessed on 6 March 2017) (FDR ≤ 0.05) to identify potential metabolic pathways significantly enriched in the DEGs. All raw data were uploaded onto NCBI SRA under the sample accession number PRJNA824941.

### 2.9. Statistical Analysis

Additional statistical analysis including one-way analysis of variance (ANOVA) and two sample *t*-tests were performed using the software SPSS 20.0 (IBM Corporation, New York, NY, USA). All statistical tests performed in this study were considered significant at *p* < 0.05.

## 3. Results

### 3.1. Bacterial Wilt Disease Incidence in Experiment I

In the experiment I, the early inoculation of the pathogen resulted in significant differences in bacterial wilt disease incidences between treatments (Figure 2a) 7 days after transplanting. In detail, both treatments that were inoculated with the pathogenic *R. solanacearum* (SBF-Rs and SCF-Rs) displayed symptoms of the disease, albeit lower for SBF-Rs (53.33%) as compared to SCF-Rs (86.67%). When comparing control treatments, that is, non-inoculated plants, the SBF showed no disease incidence, while the SCF resulted in an incidence of 13.33%. Concerning the pathogen abundance, there was no significant difference between the CF and BF soils with or without pathogen inoculation (*t*-test, *P*_SBF vs. SCF_ > 0.05, *P*_SBF-Rs vs. SCF-Rs_ > 0.05). However, the pathogen abundance in the rhizosphere increased significantly after the inoculation in both soils (*t*-test, *P*_SBF vs. SBF-Rs_ < 0.05, *P*_SCF vs. SCF-Rs_ < 0.05; Figure 2b).

### 3.2. Bacterial Community Composition in Experiment I

Pathogen inoculation significantly decreased bacterial diversity (Shannon values) in BF but not in CF soil (*t*-test, *P*_SBF vs. SBF-Rs_ < 0.05, *P*_SCF vs. SCF-Rs_ > 0.05; Figure 2c). No significant difference in Chao1 values was observed between BF soil with or without pathogen inoculation (*t*-test, *P*_SBF vs. SBF-Rs_ > 0.05), but higher values were found in CF soil with *R. solanacearum* inoculation (*t*-test, *P*_SCF vs. SCF-Rs_ < 0.05) (Appendix A). NMDS based on a Bray–Curtis dissimilarity matrix and PERMANOVA analysis revealed a clear effect of *R. solanacearum* inoculation on soil bacterial communities, both in BF and CF soils (Figure 2d).

### 3.3. Responsive Bacterial Taxa in Experiment I

Overall, the numbers and relative abundance of genera with significant negative correlation with the relative abundance of *R. solanacearum* were always higher in BF than CF soils with and without pathogen inoculation (SBF-Rs: 52 genera and relative abundance of 0.0342% > SCF-Rs: 37 genera and relative abundance of 0.0304%; SBF: 43 genera and relative abundance of 0.0432% > SCF: 6 genera and relative abundance of 0.0028%; spearman’s rank correlation analysis *p* < 0.05, Figure 2e and Appendix A). When inoculated with the pathogen, the number of genera and relative abundance of correlation mostly kept constant in BF soil, and increased in CF soil (SCF-Rs > SCF, Figure 2e and Appendix A).

### 3.4. R. solanacearum Abundance in Experiment II

In Experiment II, after the pathogen inoculation, the abundance of *R. solanacearum* changed significantly (*p* < 0.05). The pathogen abundance in the rhizosphere reduced after the inoculation in both soils, however, a significant difference was observed within BF treatments (*P*_EBF vs. EBF-Rs_ < 0.05). Last, an overall significant lower abundance of *R. solanacearum* was observed in BF than in CF soil with or without pathogen inoculation (*P*_EBF-Rs vs. ECF-Rs_ < 0.05, *P*_EBF vs. ECF_ < 0.05, Figure 3a and Appendix A).

### 3.5. Bacterial Community Analysis in Experiment II

No significant difference based on Shannon and Chao1 values was observed across treatments (Figure 3b and Appendix A). The NMDS based on a Bray–Curtis dissimilarity matrix and PERMANOVA analysis showed significant variations in community composition based on soil types, but not due to pathogen inoculation (Figure 3c). Mantel test further revealed the overall bacterial community composition to significantly correlate with the pathogen suppression (Figure 3d). LEfSe analysis identified significant OTUs that differed between treatments with pathogen inoculation (EBF-Rs versus ECF-Rs). For instance, a total of 139 OTUs were enriched in EBF-Rs, and 68 OTUs were enriched in ECF-Rs (Figure 3e). Spearman’s rank correlation analysis revealed that OTUs enriched in EBF-Rs had negative correlations with the relative abundance of *R. solanacearum*, while OTUs enriched in ECF-Rs had positive correlations with the abundance of the pathogen (FDR < 0.05) (Appendix A).

### 3.6. Transcriptome Analysis in Experiment II

RNA sequencing was used to examine the gene expression patterns of *Solanum lycopersicum* in two treatments with the pathogen inoculation (EBF-Rs vs. ECF-Rs). A total of 36 million raw reads were obtained for each sample, resulting in ~98% of clean reads (Appendix A). Approximately 78% of the reads were mapped onto the *Solanum lycopersicum* genome, and 98% were uniquely mapped (Appendix A).

Between EBF-Rs and ECF-Rs treatments, within the total 33,785 genes, 892 genes were found to be differentially expressed (DEGs). In EBF-Rs, 671 genes were up-regulated and 221 genes were down-regulated in comparison with ECF-Rs (|fold change| > 2, *p*-value <0.05, Figure 4a). Gene Ontology enrichment analysis showed that compared with ECF-Rs, the enriched DEGs were annotated within Cellular Component and Biological Process, including cellular component organization, fruit ripening, metabolic process, response to stress, sequence-specific DNA binding transcription factor activity, cytoskeleton, external encapsulating structure, extracellular region and membrane obtained (*p* < 0.05 Figure 4b). The KEGG analysis revealed the metabolism category as the most affected pathway in EBF-Rs. Within that, the carbohydrate metabolism, amino acid metabolism, metabolism of terpenoids and polyketides, and biosynthesis of other secondary metabolites were significantly enriched (Appendix A). The KEGG pathway enrichment analysis revealed the zeatin biosynthesis, thiamine metabolism, steroid biosynthesis, starch and sucrose metabolism, phenylpropanoid biosynthesis, pentose and glucuronate interconversions, carotenoid biosynthesis, and C5−Branched dibasic acid metabolism to be more expressed in EBF-Rs (*p* < 0.05, Figure 4c).

### 3.7. Relationship between Microbial Community and Gene Expression

Mantel test revealed a significant positive correlation between the bacterial community and the phenylpropanoid biosynthesis pathway (*p* = 0.00426, R^2^ = 0.479, Appendix A). Network analysis was further used to determine the relationships between genes in the phenylpropanoid biosynthesis pathway and enriched OTUs (EBF-Rs and ECF-Rs). All 143 genes in phenylpropanoid biosynthesis pathway and all enrichment OTUs in EBF-Rs (139) and ECF-Rs (68) were subjected to network analysis (Figure 5a). The results showed that there were 120 connections between genes and EBF-Rs enriched OTUs, with 80.83% positive connections, while the connections between genes and ECF-Rs enriched OTUs were only 13 and most connections were negative (69.23%). A total of 17 OTUs with positive connection with DEGs were further observed, all these OTUs were enriched in EBF-Rs and connected with up-regulated genes. In addition, their respective relative abundances in BF soil were significantly higher than that in CF soil (EBF-Rs > ECF-Rs, EBF > ECF). Besides, 3 OTUs with negative connections with DEGs were detected, all of which were enriched in ECF-Rs and connected with up-regulated genes. Their respective relative abundances in CF soil were significantly higher than in BF soil (EBF-Rs < ECF-Rs, EBF< ECF) (Figure 5b). We defined OTUs that positively correlated with up-regulated genes as “enriched group”, and those that negatively correlated with up-regulated genes as “depleted group”. The taxonomic affiliation of these OTUs revealed the enriched group to be composed of members belonging to *Gp6* (OTU30 and OTU1320), *Actinomarinicola* (OTU917), *Niastella* (OTU1693), *Phaeodactylibacter* (OTU201), *Longimicrobium* (OTU253), *Bythopirellula* (OTU444 and OTU3582), *Brevundimonas* (OTU1029 and OTU4576), *Ferruginivarius* (OTU873), *Kushneria* (OTU282), *Methylomarinovum* (OTU626), *Pseudolabrys* (OTU764), *Sphingobium* (OTU276), *Sphingomonas* (OTU5334) and *Alterococcus* (OTU859), and the depleted group mostly affiliated with *Arenimonas* (OTU290), *Luteitalea* (OTU278) and *Patulibacter* (OTU236) (Figure 5b).

The combined analysis of both experiments revealed the consistent relative abundance of members of the enriched group to be higher in BF than in CF soil with or without the pathogen inoculation (EBF-Rs > ECF-Rs, EBF > ECF, SBF-Rs > SCF-Rs, SBF < SCF). In the case of pathogen inoculation, these members increased in BF and decreased in CF soil (EBF-Rs > EBF, SBF-Rs > SBF, ECF-Rs < ECF, SCF-Rs < SCF). Conversely, the members of the depleted group were found to be consistently higher in CF than in BF soil with or without the pathogen inoculation (EBF-Rs < ECF-Rs, EBF < ECF, SBF-Rs < SCF-Rs, SBF > SCF). Furthermore, in the case of pathogen inoculation, these members decreased in BF and increased in CF soil in the second experiment (EBF-Rs < EBF, SBF-Rs < SBF, ECF-Rs > ECF, SCF-Rs < SCF) (Figure 5c).

## 4. Discussion

Soilborne diseases represent economically important threats to several crops, including tomato, watermelon, and banana [11,19,35,36]. Several studies have been focusing on the characterization of microbial communities in disease-suppressive soils and how these relate to the establishment of specific disease-suppressive microbial taxa in the plant rhizosphere [1,7,13]. Despite the phenomenon of disease suppression can be linked with the ability of local soil and rhizosphere microbes to antagonize pathogens—or at least to keep their population size under control—knowledge of how specific microbes modulate plant systemic resistance remains still largely elusive. This holds great promise for advancing our understanding of the mechanisms by which specific microbial taxa and their association with plants lead to the activation of plant immunity defense.

Our first experiment tested the tomato wilt disease incidence once the pathogen was inoculated at the same time seedlings were transferred to the split-root system. This was carried out for soils historically displaying different levels of disease incidence. In sum, we found the soil with a legacy of suppressiveness to display consistently significantly lower disease incidence regardless of the time at which the pathogen was inoculated (i.e., at the time of seedling transplant or 7 days after). However, we found the abundance of the pathogen in the rhizosphere to increase significantly both in the suppressive and conducive soils after the pathogen inoculation. Previous studies corroborate this finding of pathogen accumulation throughout plant growth, including studies performed on tomato and *Arabis alpina* [7,16]. Moreover, in the second experiment when the pathogen inoculation was performed 7 days after the seedling transplant, the abundance of *R. solanacearum* displayed a significant decrease in the suppressive soil when compared with the conducive soil. This likely relates to the fact that the provision of an initial time for the establishment of the rhizosphere microbiome prior to pathogen exposure directly influences pathogen suppression in suppressive soils. Besides, it is well-known that the plant rhizosphere microbiome tends to be different when plants are grown in disease-suppressive or conducive soils [1,7,8], and several beneficial rhizobacterial genera like *Pseudomonas* [7,17], *Bacillus* [35,37], *Paenibacillus* [38], and *Streptomyces* [39,40] are often identified as disease-suppressive taxa.

Our root transcriptome analysis revealed significant differences in the gene expression patterns of tomato plants cultivated in suppressive and conducive soils. Previous studies support this notion of changes in plant transcriptional patterns due to differences in soil type and microbiome composition. For example, Jones (2006) [41] and Dodds (2010) [42] reported that plants were able to shift the transcription when exposed to pathogens. Our analysis revealed the phenylpropanoid biosynthesis pathway to be enriched in roots from the suppressive soil compared with the plants from the conducive soil. Phenylpropanoids play an important role in plant growth and development, and several important secondary metabolites (e.g., flavonoids, and salicylic acid) are produced and metabolized within this pathway [43,44,45]. Previous studies have shown the phenylpropanoid biosynthesis to be correlated with induced resistance in response to biotic and abiotic stresses, such as pathogens infection, insect herbivory, UV irradiation, and low temperature [46,47,48,49].

We also identified significant ‘enriched’ bacterial taxa associated with upregulated genes in the phenylpropanoid biosynthesis pathway. This might likely relate to the potential of these microbial taxa to induce plant systemic resistance. In detail, the bacterial taxa significantly correlated with the expression of resistance genes (DEGs in the phenylpropanoid biosynthesis pathway) included members within the *Gp6* (OTU30 and OTU1320), *Actinomarinicola* (OTU917), *Niastella* (OTU1693), *Phaeodactylibacter* (OTU201), *Longimicrobium* (OTU253), *Bythopirellula* (OTU444 and OTU3582), *Brevundimonas* (OTU1029 and OTU4576), *Ferruginivarius* (OTU873), *Kushneria* (OTU282), *Methylomarinovum* (OTU626), *Pseudolabrys* (OTU764), *Sphingobium* (OTU276), *Sphingomonas* (OTU5334), and *Alterococcus* (OTU859). Our results are partially supported by a previous study in postharvest citrus, showing that *Pichia galeiformis* can reduce the disease incidence of the pathogen *Penicillium digitatum* via induced resistance by triggering the phenylpropanoid biosynthesis pathway [50]. In addition, a large number of studies have previously reported correlations between microbial taxa and defense signaling in plants. For instance, in *Arabidopsis*, the relative abundance of *Firmicutes* in the rhizosphere was positively correlated with the plant immune response, and the deletion of SA or JA signaling genes was found to significantly affect the rhizosphere microbiome [15,51].

It is worth noticing that our analysis also identified several other bacterial taxa potentially associated with plant health. For example, the enrichment of the seed-endophytic bacterium *Sphingomonas* was shown to produce anthranilic acid and induce resistance in disease-susceptible rice genotypes [52]. In our previous study, three *Sphingomonas* strains isolated from the tomato rhizosphere also showed to be effective in the control of bacterial wilt disease [24]. Furthermore, members of the genus *Niastella* were reported to be important biocontrol agents against stripe rust in wheat [53]. Conversely, *Sphingobium* spp. was reported to contribute to the suppression of banana *Fusarium* wilt disease [54,55]. Furthermore, members within the *Gp6* were reported to occur at higher relative abundances in the rhizosphere of strawberry-resistant cultivars and in banana *Fusarium* wilt disease-suppressive soils [56,57]. Some studies also described that species belonging to the genus *Brevundimonas* can act as plant growth-promoting rhizobacteria and/or act as bioremediation agents [58,59]. Last, many other bacterial taxa reported in this study (including *Actinomarinicola, Phaeodactylibacter*, *Longimicrobium*, *Bythopirellula*, *Ferruginivarius*, *Kushneria*, *Methylomarinovum, Pseudolabrys,* and *Alterococcus*) have not yet been reported in terms of their mechanisms potentially associated with plant-disease control. These taxa can be further explored in terms of their interactions in the rhizosphere and potential mechanisms associated with antagonism against pathogens and/or induction of plant systemic resistance.

## Figures and Tables

**Figure 1 plants-11-02816-f001:**
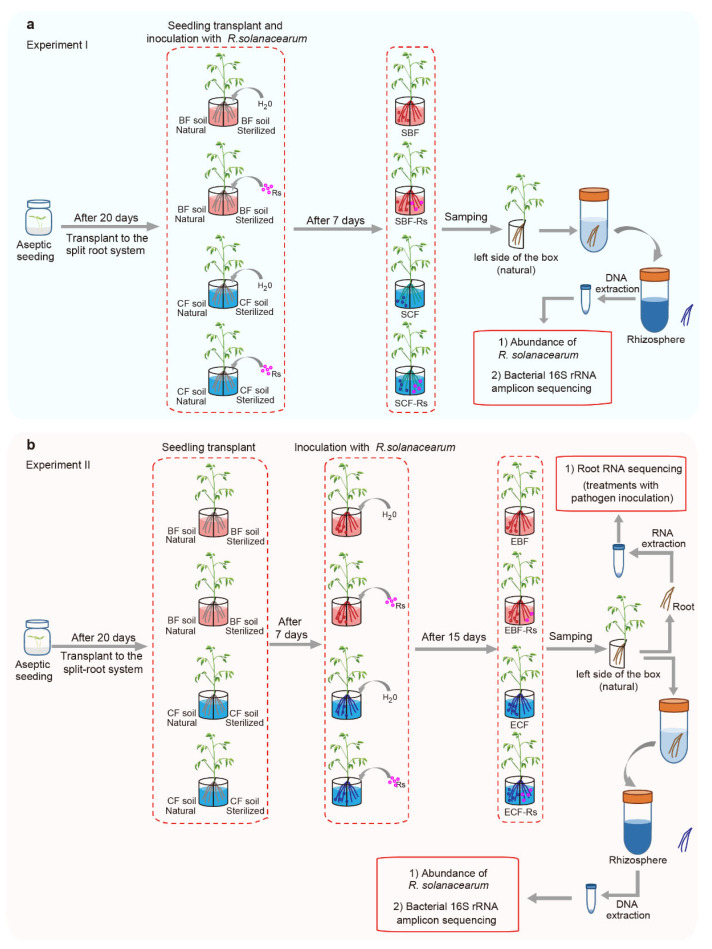
Schematic representation of the experimental systems used for the two independent experiments. Experiment I (**a**): the pathogen was inoculated at the same time that the seedlings were transferred to the split-root system (S); SBF: rhizosphere derived from the BF soil without *R. solanacearum* inoculation; SCF: rhizosphere derived from the CF soil without *R. solanacearum* inoculation; SBF-Rs: rhizosphere derived from the BF soil and inoculated with *R. solanacearum* at the same time that plants were transferred to the split-root system; SCF-Rs: rhizosphere derived from the CF soil and inoculated with *R. solanacearum* at the same time that plants were transferred to the split-root system. Experiment II (**b**): the pathogen was inoculated 7 days after the seedlings were transferred to the split-root system (E). EBF: rhizosphere derived from the BF soil without *R. solanacearum* inoculation; ECF: rhizosphere derived from the CF soil without *R. solanacearum* inoculation; EBF-Rs: rhizosphere and root derived from the BF soil and inoculated with *R. solanacearum* 7 days after the plant establishment in the split-root system; ECF-Rs: rhizosphere and root derived from the CF soil and inoculated with *R. solanacearum* 7 days after the plant establishment in the split-root system.

**Figure 2 plants-11-02816-f002:**
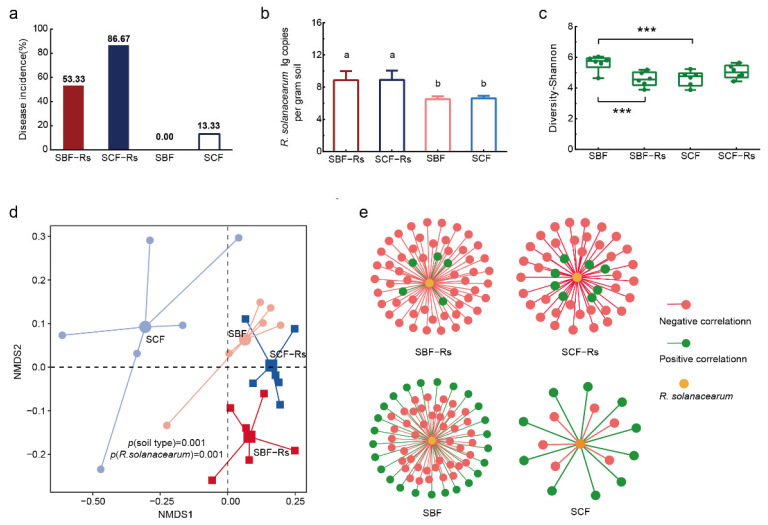
Bacterial wilt disease incidence, bacterial community composition and responsive bacterial taxa in Experiment I. (**a**) Bacterial wilt disease incidence (%); (**b**) The abundance of *R. solanacearum* in the rhizosphere obtained via quantitative PCR; (**c**) *Alpha* diversity analysis of rhizosphere bacterial communities (Shannon index); (**d**) Non-metric multidimensional scaling (NMDS) and PERMANOVA analysis of rhizosphere bacterial communities; (**e**) Analysis of responsive microbial taxa. Network showing the number of significant responsive bacterial genera. SBF: rhizosphere derived from the BF soil without *R. solanacearum* inoculation; SCF: rhizosphere derived from the CF soil without *R. solanacearum* inoculation; SBF-Rs: rhizosphere derived from the BF soil and inoculated with *R. solanacearum* at the same time that plants were transferred to the split-root system; SCF-Rs: rhizosphere derived from the CF soil and inoculated with *R. solanacearum* at the same time that plants were transferred to the split-root system. Different letters above the bars indicate significant differences at *p* < 0.05 based on one-way analysis of variance (ANOVA). The asterisks indicate significant differences between treatments determined by two-sample *t*-tests (*** *p* < 0.001).

**Figure 3 plants-11-02816-f003:**
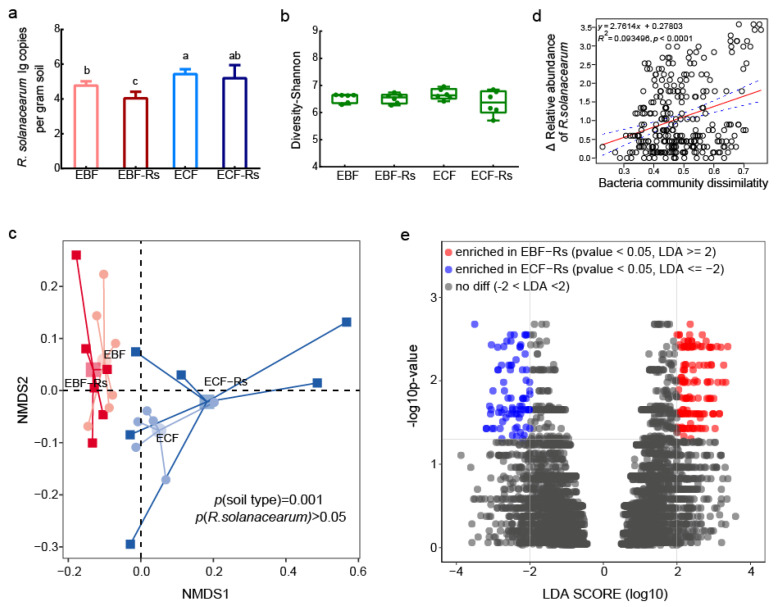
*R. solanacearum* abundance and bacterial community analysis of Experiment II. (**a**) The abundance of *R. solanacearum* in the rhizosphere obtained via quantitative PCR; (**b**) *Alpha* diversity analysis of rhizosphere bacterial communities (Shannon index); (**c**) Non-metric multidimensional scaling (NMDS) and PERMANOVA analysis of rhizosphere bacterial communities; (**d**) Mantel test used to identify the correlation between bacterial communities with relative abundance values of *R. solanacearum*; (**e**) LEfSe analysis displaying significant differences in community composition between treatments with pathogen inoculation (LDA ≥ 2 and *p* < 0.05). EBF: rhizosphere derived from the BF soil without *R. solanacearum* inoculation; ECF: rhizosphere derived from the CF soil without *R. solanacearum* inoculation; EBF-Rs: rhizosphere derived from the BF soil and inoculated with *R. solanacearum* 7 days after the plant establishment in the split-root system; ECF-Rs: rhizosphere derived from the CF soil and inoculated with *R. solanacearum* 7 days after the plant establishment in the split-root system. Different letters above the bars indicate significant differences at *p* < 0.05 based on one-way analysis of variance (ANOVA).

**Figure 4 plants-11-02816-f004:**
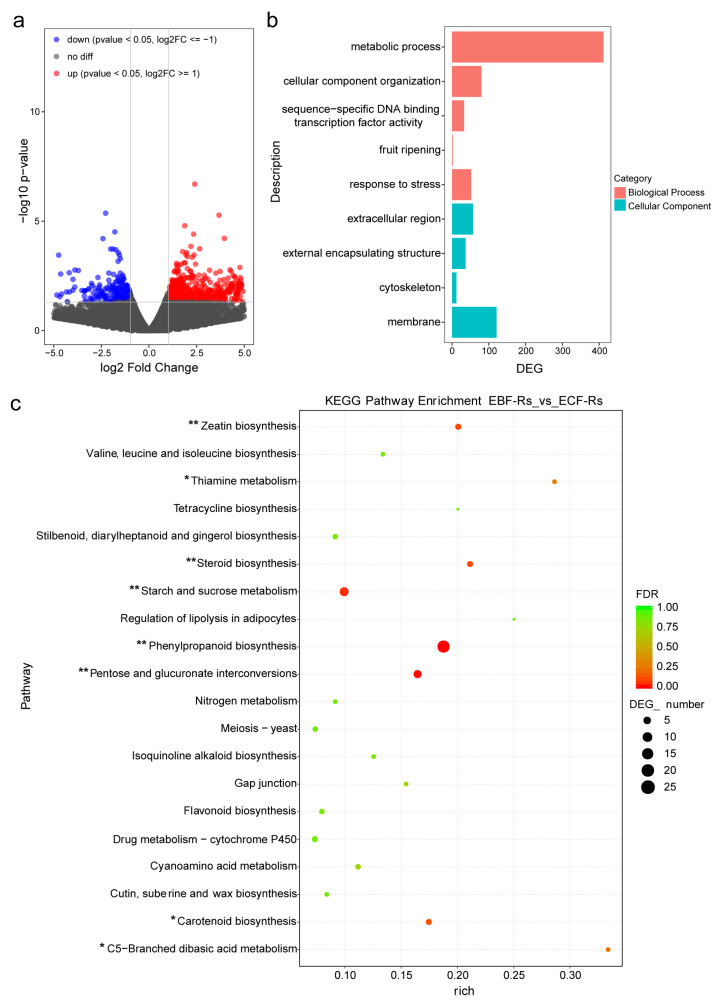
Transcriptome analysis of Experiment II. (**a**) Volcano diagram displaying the differentially expressed genes (DEGs, |log2fold-change| > 1.0 and *p*-value < 0.05); (**b**) Gene Ontology enrichment analysis (FDR ≤ 0.05); (**c**) KEGG pathway enrichment analysis (FDR ≤ 0.05). EBF-Rs: root derived from the BF soil and inoculated with *R. solanacearum* 7 days after the plant establishment in the split-root system; ECF-Rs: root derived from the CF soil and inoculated with *R. solanacearum* 7 days after the plant establishment in the split-root system. The asterisks indicate significant differences between treatments (* *p* < 0.05, ** *p* < 0.01).

**Figure 5 plants-11-02816-f005:**
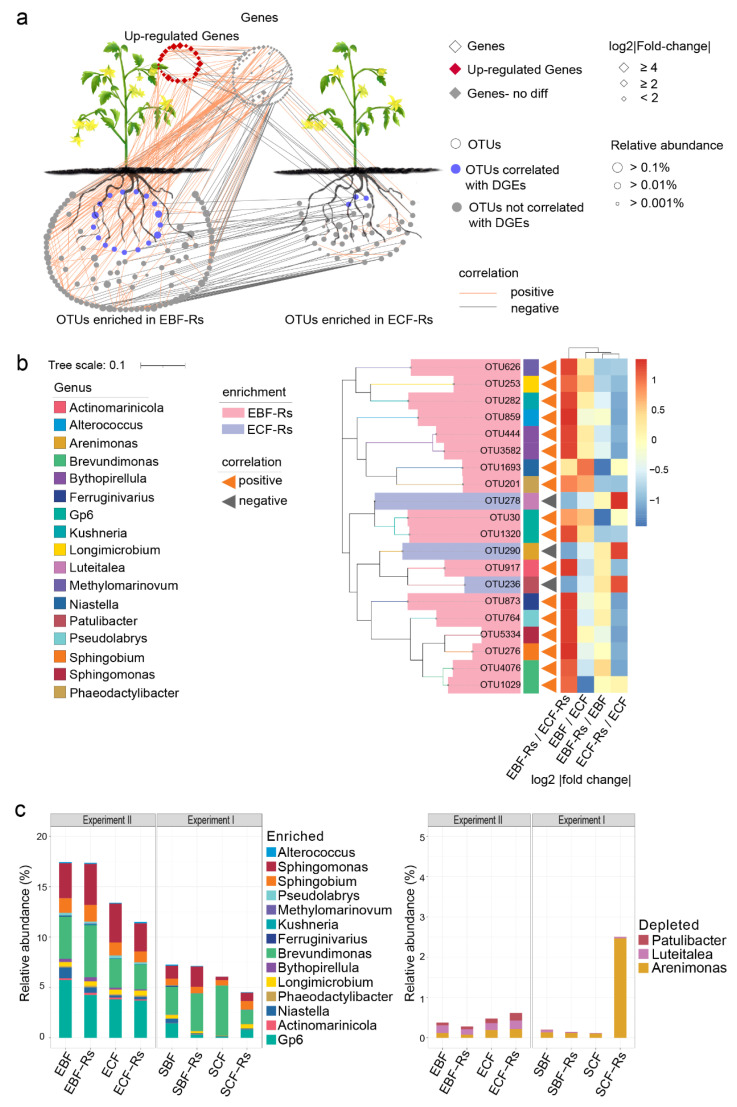
Correlational analysis between microbial taxa and root gene expression. (**a**) Network analysis displays all possible correlations between OTU × OTU, OTU × gene, and gene × gene (all 143 genes in phenylpropanoid biosynthesis pathway and all enrichment OTUs in EBF-Rs (139) and ECF-Rs (68), Spearman’s correlation coefficient (*r*) > 0.6 and statistically significant (*p*-value < 0.05); (**b**) OTUs significantly correlated with DEGs, the heatmap displays the relative change (fold change) of OTUs between the two treatments; (**c**) Relative abundances of specific genera in the two experiments. Enriched group: OTUs that positively correlated with up-regulated genes; depleted group: OTUs that negatively correlated with up-regulated genes. In experiment II, EBF: rhizosphere derived from the BF soil without *R. solanacearum* inoculation; ECF: rhizosphere derived from the CF soil without *R. solanacearum* inoculation; EBF-Rs: rhizosphere and root derived from the BF soil and inoculated with *R. solanacearum* 7 days after the plant establishment in the split-root system; ECF-Rs: rhizosphere and root derived from the CF soil and inoculated with *R. solanacearum* 7 days after the plant establishment in the split-root system. In experiment I, SBF: rhizosphere derived from the BF soil without *R. solanacearum* inoculation; SCF: rhizosphere derived from the CF soil without *R. solanacearum* inoculation; SBF-Rs: rhizosphere derived from the BF soil and inoculated with *R. solanacearum* at the same time that plants were transferred to the split-root system; SCF-Rs: rhizosphere derived from the CF soil and inoculated with *R. solanacearum* at the same time that plants were transferred to the split-root system.

## Data Availability

All raw data were uploaded onto NCBI SRA under the sample accession number PRJNA824941.

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
