# Peer review of "Partitioning the Effects of Soil Legacy and Pathogen Exposure Determining Soil Suppressiveness via Induced Systemic Resistance"

_plants, 2022, doi:10.3390/plants11212816_

Round 1

Reviewer 1 Report

Dear Sirs,

I have much appreciated your work for its abundance of analyses and its advanced statistical and bioinformatic background. In my opinion the findings are quite significant, since some bacterial species, associated with an increased expression of genes involved in the biosynthesis pathways of phenylpropanoid, are identified and suggested for further investigations.

I only found a little confusing the distribution of figures between main paper and supplementary material, and in general I perceived them as slightly redundant. Maybe something in this sense can be improved. For the rest, I made some minor notes in the attached file.

I recommend its acceptation after a minor revision.

Best regards

Author Response

General comments

I have much appreciated your work for its abundance of analyses and its advanced statistical and bioinformatic background. In my opinion the findings are quite significant, since some bacterial species, associated with an increased expression of genes involved in the biosynthesis pathways of phenylpropanoid, are identified and suggested for further investigations.

I only found a little confusing the distribution of figures between main paper and supplementary material, and in general I perceived them as slightly redundant. Maybe something in this sense can be improved. For the rest, I made some minor notes in the attached file.

I recommend its acceptation after a minor revision.

Response:

We thank the reviewer for all comments and suggestions provided throughout our manuscript. Please, see our point-by-point responses below (in bold).

-Possible redundancy across figures was checked and corrected accordingly.

[Comment 1]. Line 67 - The SA signaling, however, is associated more properly with the so-called Systemic Aquired Resistance (SAR).

Response:

Correction made. Now it reads “For instance, the microbial-activation of jasmonic acid (JA) and ethylene (ET)-dependent elicitation were recently reported in Arabidopsis and tomato plants [22,23]”.

[Comments 2]. Line 73 - Please add the species authors, since here it is mentioned for the first time: Ralstonia solanacearum (Smith) Yabuuchi et al. emend. Safni et al.

Response:

Correction made. Now it readsRalstonia solanacearum (Smith) Yabuuchi et al. emend. Safni et al” in the main text.

[Comments 3]. Line 84 - Why such (BF and CF,)? How are they derived?

Response:

This issue was corrected. We modified the position where these abbreviations firstly appeared in the main text.

[Comments 4]. Lines 86-87 - You should at least briefly summarize this method of suppressiveness assessment.

Response:

Additional information was added in the main text. Now it reads “The disease suppressiveness of the suppressive soil was determined in our previous study [24]. In brief, conducive soils were mixed with 10% of suppressive soils heat-treated at different temperatures. Then, the disease suppressiveness was evaluated via seedling disease incidence”.

[Comments 5]. Line 93 - Please, remind in this section that the specific disease is induced by R. solanacearum. Besides, the correct disease name is bacterial wilt.

Response:

Correction made.

[Comments 6]. Line 101 - Add its author: Solanum lycopersicum L.

Response:

Information added.

[Comments 7]. Line 102 - Again, briefly summarize this method of sterilization.

Response:

Information was added in the main text. Now it reads “Seeds were surface-sterilized using 70% ethanol for 10min, 100% ethanol for 1min, and 3% NaClO for 1min”.

[Comments 8]. Lines 147-160 - How are derived the acronyms SBF and EBF?

Response:

We defined these acronyms in the legend of Figure 1. Now it reads “the pathogen was inoculated at the same time that the seedlings were transferred to the split-root system (S)”; “the pathogen was inoculated 7 days after the seedlings were transferred to the split-root system (E)”.

Reviewer 2 Report

The revised manuscript studies important topic about soil quality affecting plants growth. The presence of valuable microbiota in the rhizosphere often improve plant growth. This topic is important in agriculture but also environmental sciences. This is emphasized in introduction. As Plants Journal aims on such topic, including physiological aspects, coping with stressors, this manuscript perfectly falls into its scope.

The title is interesting, abstract presents immediately interesting outcome and suggest the use of modern molecular methods. Keywords are adequate to this topic and literature too.

Introduction is a good screening for given topic and contains the rationale for this study with clear aim and hypothesis.

Experiment is perfectly designed and conducted. Figure 1 presents clearly the whole design. Description is clear and the experiment may by easily reproduced. Modern molecular and bioinformatic methods were applied supported with proper statistics what allows to believe that data are genuine. These results are very well described with statistical results, what is not so common. Many graphs (including supplementary) are helpful and have good quality. Discussion supports results very well, all leading to convincing conclusions.

Additional comments:

1. Line 51: please add space before '[11]'

2. What was the number of samples collected and their mass?

3. What was the origin of pathogen?

4. Is Table S1 cited anywhere?

Author Response

General comments
The revised manuscript studies important topic about soil quality affecting plants growth. The presence of valuable microbiota in the rhizosphere often improves plant growth. This topic is important in agriculture but also environmental sciences. This is emphasized in introduction. As Plants Journal aims on such topic, including physiological aspects, coping with stressors, this manuscript perfectly falls into its scope.
The title is interesting, abstract presents immediately interesting outcome and suggest the use of modern molecular methods. Keywords are adequate to this topic and literature too.
Introduction is a good screening for given topic and contains the rationale for this study with clear aim and hypothesis.
Experiment is perfectly designed and conducted. Figure 1 presents clearly the whole design. Description is clear and the experiment may by easily reproduced. Modern molecular and bioinformatic methods were applied supported with proper statistics what allows to believe that data are genuine. These results are very well described with statistical results, what is not so common. Many graphs (including supplementary) are helpful and have good quality. Discussion supports results very well, all leading to convincing conclusions.
Response:
We thank the reviewer for the positive feedback, and for all comments and suggestions provided throughout the manuscript. Please, see our point-by-point responses below (in bold).

Additional comments:
[Comments1]. Line 51: please add space before '[11]'.
Response:
Correction made.

[Comments2]. What was the number of samples collected and their mass?
Response:
Information was added in the main text. Now it reads “Rhizosphere samples were collected using six samples in each treatment, and the mass of each sample was greater than 0.5g.”; “Roots samples were collected using three samples in each treatment, and the mass of each sample was greater than 2.0 g”.

[Comments3]. What was the origin of pathogen?
Response:
Information was added in the main text. Now it reads “isolated from the rhizosphere of a wilted plant in an infested field site in Qilin Town (118°57′E, 32°03′N, Nanjing city, China)”.

[Comments4]. Is Table S1 cited anywhere?
Response:
Correction made. This table is now cited in the following statement “A total of 36 million raw reads were obtained for each sample, resulting in ~98% of quality-filtered reads (Table S1)”.
